# Experimental Study on the Physical and Mechanical Characteristics of Refractory Concrete Using Heat-Treated Steel Slag Coarse Aggregates

**Munaf Alkhedr [1], Majed Asaad [1], Mahmoud Ismail [1] and George Wardeh [2],\***

[1] Department of Transportation and Structural Materials, Faculty of Civil Engineering, Damascus University, Damascus P.O. Box 30621, Syria; munaf.alkhedr@damascusuniversity.edu.sy (M.A.); majaasaad@damascusuniversity.edu.sy (M.A.); mahmoud.ismail@damascusuniversity.edu.sy (M.I.)

[2] L2MGC, CY Cergy-Paris University, 95031 Neuville-sur-Oise, France

\* Correspondence: george.wardeh@cyu.fr; Tel.: +33-(0)-134256828

**Abstract:** The aim of this study is to compare the properties of refractory concrete made with thermally treated and untreated steel slag. Five concrete compositions were prepared and investigated in the present work. The first mixture, referred to as the reference, was formulated using dolomite aggregates, whereas the second and third mixtures were developed by replacing natural coarse aggregate with 50 and 100% by weight of thermally untreated steel slag, respectively. The same replacement ratio (50% and 100%) of thermally treated steel slag was used to produce the fourth and fifth mixtures. Specimens of each specimen were placed in a furnace and heated to 400 °C and 800 °C. The mass loss for all the specimens heated to 400 °C was about 8%, while the reference suffered the maximum mass loss at 800 °C, which was 21.6%. The mixture with a 100% substitution of thermally treated steel slag produced the maximum compressive strength when compared to other mixtures at a temperature of 800 °C. The compressive strength of the M5 mixture was 18 MPa versus 10.87 MPa for the reference mixture. Additionally, optical microscope examination of specimens containing thermally treated steel slag revealed less damage than that observed in mixtures with dolomite.

**Keywords:** steel slag; thermal treatment; mechanical properties; refractory concrete





## 1. Introduction

For all nations to contribute to sustainable development, the use of waste in concrete and cement products has become a must [1]. There have been numerous materials employed, including recycled aggregates [2,3], olive stones as a lightweight aggregate in mortars [4], steel slag as a coarse aggregate in concrete, and mortars doped with crumb rubber from used tires [5]. Steel slag is a by-product of the steel industry and is a type of waste. According to a certain type or grade of steel and the furnace used during the steel-making process, many forms of steel slag are produced [6]. Steel slag can be estimated to account for 10% to 15% of the world's 1878.5 million tons of crude steel produced in December 2022 [7–10].

Steel slag should be recycled and transformed into new applications because landfill disposal of this material could have a negative impact on the environment. Indeed, the majority of the industrialized countries in the world use steel slag in a variety of engineering, agricultural, and business applications. The recycling rate differs from nation to nation; it is 70% in the USA [10], 72.4% in Europe [11], 98% in Japan [12], and barely 30% in China [13,14].

Numerous studies have examined the use of steel slag as a coarse aggregate in concrete mixes and for road construction, in addition to the fact that it is now widely used in mining, filtration systems, the construction of roads, and the fertilizer sector [15,16].

Slag-containing concrete has acceptable mechanical properties and volume changes for structural application, but it might pose a risk for corrosion of the reinforcement if the slag has high sulfur content [17]. Steel slag concrete is less workable than conventional concrete due to its high water absorption capacity in the fresh state. However, after 90 days of curing, it demonstrates extremely good compressive strength improvements of 18%. The tensile strength of normal concrete also increases when steel slag is added [18].

In terms of durability, slag could improve the fire performance of concrete in the range of temperatures up to 400 °C, and this improvement is more obvious in concrete made with previously thermally treated slag [19].

The properties of concrete produced with steel slag as the coarse aggregate are impacted by the aging process of these aggregates. When combined with chemical and mineral admixtures, steel slag can be used as an aggregate in concrete after an appropriate aging period to produce high-quality concrete. The use of steel slag improves the mechanical properties of concrete after an aging time of 30 months. This time is sufficient to complete the interactions of free calcium and magnesium oxide with water, which contributes to the volumetric stability of the steel slag [20]. As it enhances the compressive strength by 20.55% and the tensile strength by 10.55% at the replacement ratio of 40% [21], steel slag can be used as fine and coarse aggregates with specified replacement ratios.

In addition to greatly enhancing the characteristics of concrete, the microstructure of concrete including both fine and coarse steel slag exhibits complicated behavior [22]. However, to some extent, it may also be detrimental to concrete. Fine steel slag aggregate can be utilized to enhance recycled aggregate concrete [23], as well as recycled low-strength compacted concrete [24] and 3D printed concrete produced using environmentally friendly materials [24].

Steel slag's high density compared to natural materials and volume instability are the two main drawbacks that limit its usage in civil engineering, especially when concrete is exposed to high temperatures. Steel slag contains (C2S), which causes volume expansion, in four forms: ($\alpha$, $\alpha$-1, $\beta$, and $\gamma$). $\alpha$C2S is stable at temperatures above 630 °C, but $\beta$C2S begins to transform into $\gamma$C2S at temperatures above 500 °C, leading to an increase in volume expansion of up to 10% [25].

The aim of this study is to improve the mineral composition of steel slag through thermal treatment to a temperature of 1000 °C, which is intended to allow for its use it as a coarse aggregate in concrete with a positive effect on the mechanical properties of refractory concrete. To achieve this aim, an experimental program was carried out. This program includes determining the physical and mechanical properties of materials (cement, natural aggregates, and steel slag aggregate) used in this research. In addition, the properties of fresh and hardened concrete mixtures were determined. The results were compared to those of the reference mixture (without steel slag). Also, a microscopic examination was performed for all mixtures before and after exposure to high temperatures.

## 2. Materials and Methods

### 2.1. Materials

#### 2.1.1. Cement

Ordinary Portland cement (CEM I 32.5N) was used in accordance with BS EN 197-1 2011 [26], where the chemical composition is given in Table 1.

**Table 1.** Properties of Portland cement.

| Properties of Portland Cement | | | | | | | | | |
|---|---|---|---|---|---|---|---|---|---|
| Chemical composition | Constituent Content (% of mass) | CaO 59.08 | SiO$_2$ 19.12 | Al$_2$O$_3$ 5.03 | Fe$_2$O$_3$ 4.21 | MgO 2.47 | CaO$_{free}$ 1.56 | SO$_3$ 3.22 | L.O.I 1.62 |

**Table 1.** *Cont.*

| Properties of Portland Cement | | |
|---|---|---|
| Physical properties | Specific gravity | 3.15 |
| | Fineness (Blaine) m$^2$/kg | 302 |
| | Initial setting time (min) | 135 |
| Mechanical properties | Compressive strength MPa at (7 days) | 24.3 |
| | Compressive strength MPa at (28 days) | 35.6 |

### 2.1.2. Aggregates

The coarse and fine aggregates were crushed dolomite aggregates obtained from the Al-Marah area in rural Damascus. Physical property tests were carried out on the aggregates in accordance with BS EN 1097 parts 6 and 9 [27,28] and BS EN 933 parts 1 and 2 [29,30]. The particle size distributions for coarse and fine aggregates are shown in Figure 1 according to the ASTM C33 [31], ASTM C136 [32], ASTM C127 [33], and ASTM C128 [34] standards, and the physical properties are presented in Table 2.

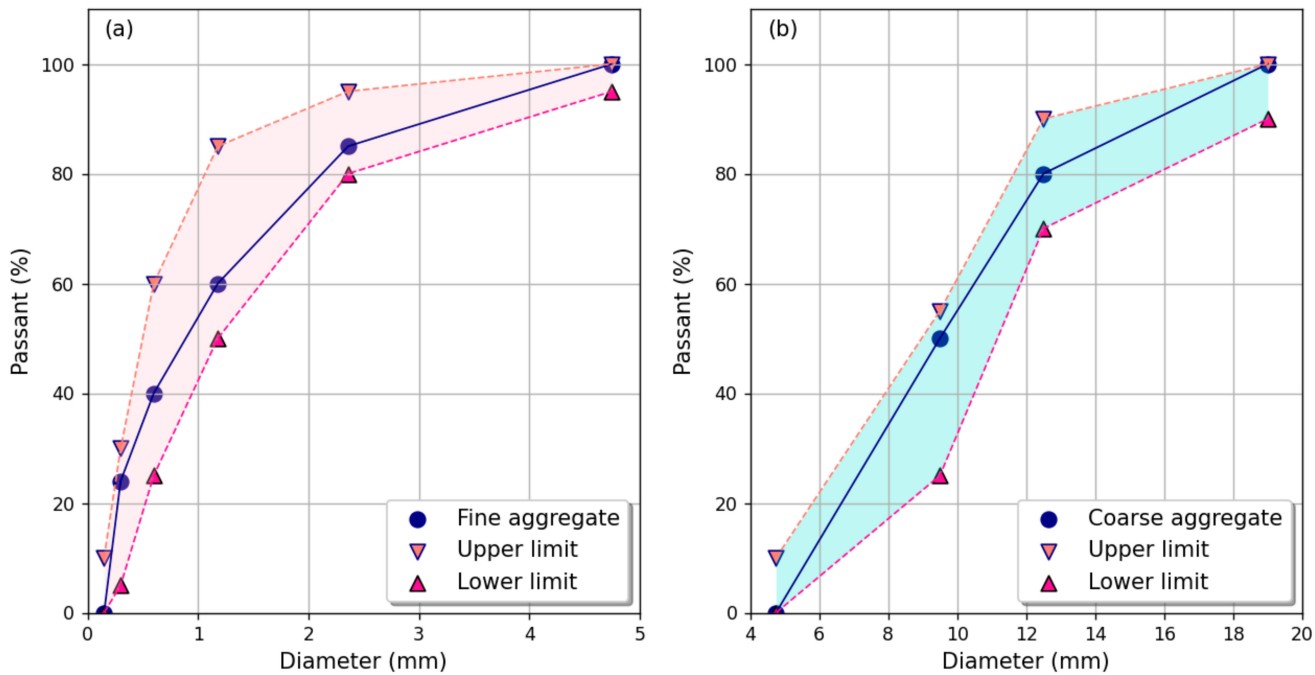

**Figure 1.** The particle size distribution for (**a**) coarse aggregate and (**b**) fine aggregate.

**Table 2.** The physical properties of natural aggregate and steel slag.

| Property | Thermally Untreated Steel Slag | Thermally Treated Steel Slag | Natural Coarse Aggregate | Natural Fine Aggregate |
|---|---|---|---|---|
| Specific gravity | 3.592 | 3.493 | 2.802 | 2.794 |
| WA (%) | 2.564 | 3.498 | 1.854 | 1.6 |
| LA coefficient (%) | 21.98 | 17.64 | 18.94 | - |
| Sand equivalent (%) | - | - | - | 75 |

### 2.1.3. Steel Slag

Air-cooled steel slag was brought in from a Syrian plant in the industrial city of Adra. The slag could not be used as a coarse aggregate due to its condition as disposable

waste and had to be prepared prior to testing. These operations included crushing and screening the material and separating it into fractions (4.75–12.5 mm, 12.5–19 mm) before then watering the fractions. The chemical composition and physical properties of steel slag are listed in Table 2. The table shows that the Los Angeles coefficient of steel slags is very close to that of natural aggregate and that the thermal treatment helped to improve this value, as it was 6.86% better than natural aggregate. In addition, steel slags have a high specific gravity compared to natural aggregate and a significantly higher water absorption rate. The chemical composition of the steel slag is given in Table 3.

**Table 3.** Chemical composition of steel slag.

| Constituent | CaO | $Fe_2O_3$ | $SiO_2$ | CuO | $AL_2O_3$ | MgO | MnO | ZnO | BrO |
|---|---|---|---|---|---|---|---|---|---|
| content (% of mass) | 27.82 | 44.15 | 13.63 | 0.04 | 7.18 | 2.75 | 2.68 | 0.66 | 0.58 |

IR spectroscopy (short for infrared spectroscopy) deals with the infrared region of the electromagnetic spectrum, i.e., light with higher wavelengths and lower frequencies than visible light. In general terms, infrared spectroscopy refers to analysis of the interaction of a molecule with infrared light. The concept of IR spectroscopy can generally be analyzed in three ways: by measuring reflection, emission, and absorption. The main use of infrared spectroscopy is to determine the functional groups of molecules, which is relevant to both organic and inorganic chemistry. This technique was applied to steel slag samples before and after heat treatment, where the targeted temperatures were (20–450–650–950) °C. The results showed that these samples are free from organic residues, where the main peaks are concentrated in the region of wave numbers less than 900 $cm^{-1}$, indicating oxygen bonds with other minerals (M-O). There are differences in the positions of the peaks in this region as a result of the deposition of oxygen in certain crystals and its transfer to other crystals as a result of the occurrence of solid-state interactions. Figure 2 shows the results of this experiment.

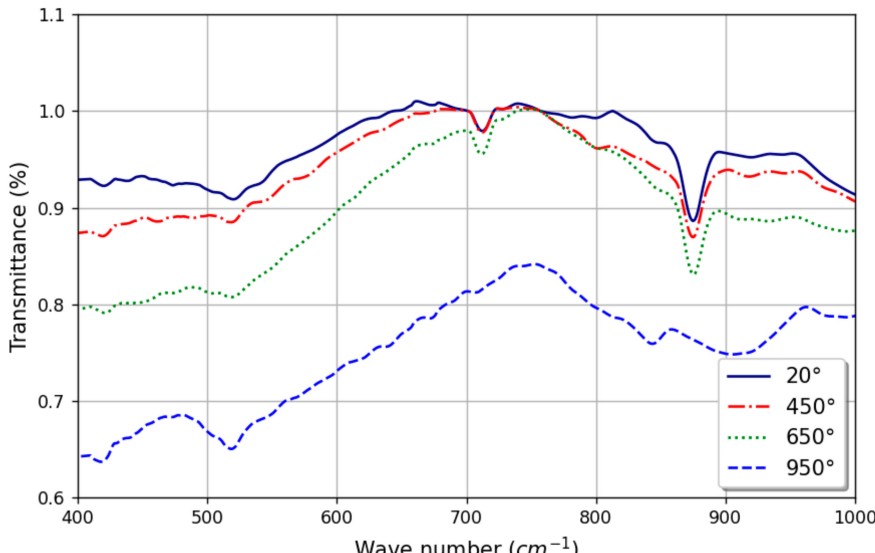

**Figure 2.** Results of IR at temperatures of (20–450–650–950) °C.

Energy Dispersive X-ray analysis (EDX), also called EDS or EDAX, is an X-ray technique for identifying material elemental composition. The data generated by EDX analysis are a set of spectra with peaks corresponding to the elements that make up the true composition of the sample being analyzed. It is also possible to carry out elemental mapping of a sample and image analysis. The elemental analysis of steel slag by EDX is shown in Table 4.

**Table 4.** The elemental analysis of steel slag according to EDX.

| Element | Fe | Ca | Si | Al | Mn | Mg | Zn | Cr | Ti | S | K | Pb | Cl |
|---|---|---|---|---|---|---|---|---|---|---|---|---|---|
| content % | 37.98 | 35.36 | 9.97 | 5.29 | 3.04 | 2.54 | 1.88 | 1.36 | 0.90 | 0.34 | 0.26 | 0.23 | 0.17 |

The method of X-ray diffraction was employed to examine the mineralogical composition of the aggregates used both before and after heat treatment. This method allows for determination of the mineral composition of the tested sample by examining the crystalline phase based on the fact that every crystal has a chemical unit that repeats frequently in accordance with a system of three axes forming meshes. These meshes are used to identify the elements since they have edges with the right lengths and angles. The sample to be examined is crushed and then spread out on a blade that revolves around an axis that lies in its plane (goniometric circle). A rotating source of monochromatic X-rays with a specific wavelength, generated by a copper anode, bombards the material. The Bragg relation, which is represented by the formula $\lambda = 2dsin\theta$, needs to be confirmed for diffraction to occur. d is the distance between the reticular planes of groups of atoms, which is intrinsic to the crystal structure; $\theta$ is the Bragg angle formed between the incident spindle and the normal to the reticular plane; and $\lambda$ is the wavelength. Slag samples were subjected to an X-ray diffraction test before and after heat treatment, and the results are shown in Figure 3 and 4, respectively. There are numerous X-ray diffraction peaks in Figure 3, and each peak denotes a different oxide among the oxides that contribute to the formation of steel slag. Figure 4 shows the production of additional peaks with large intensities following heat treatment in comparison to the material before treatment, denoting the formation of new crystalline products. Comparing the two forms before and after treatment reveals that, in accordance with the following equations, the fraction of several compounds, including C2S, C3S, and Fe, has increased:

$$SiO_2 + 2CaO \rightarrow Ca_2SiO_4$$
$$SiO_2 + 3CaO \rightarrow Ca_3SiO_5$$
$$4Fe + 3O_2 \rightarrow Fe_2O_3 \qquad (1)$$
$$CaCO_3 \rightarrow CaO + O_2$$
$$2CaO + Fe_2O_3 \rightarrow Ca_2Fe_2O_5$$

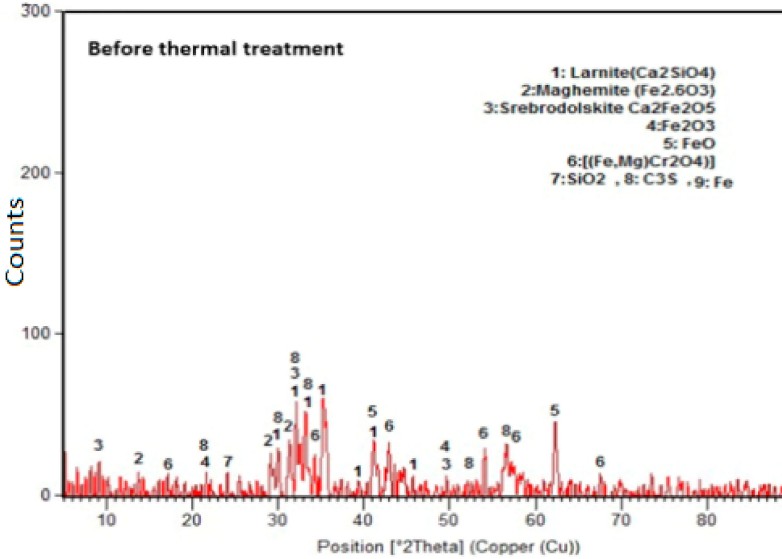

**Figure 3.** The X-ray diffraction spectrum before thermal treatment.

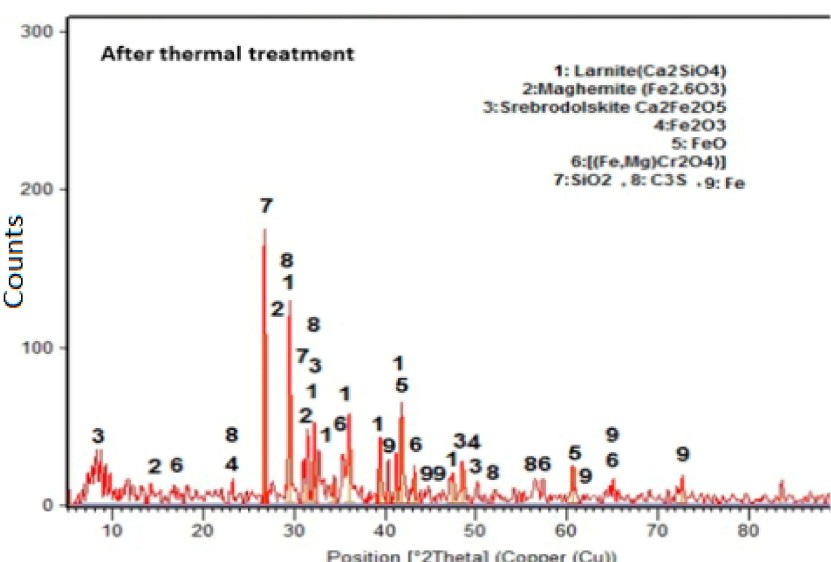

**Figure 4.** The X-ray diffraction spectrum after thermal treatment.

*2.2. Concrete Mixture Design*

According to ACI 211.1 [35], five concrete mixtures (M1, M2, M3, M4, and M5) were developed with the same cement type (CEM I 32.5N). All of the mixtures were designed to have a slump of 50 mm and an average compressive strength of about 25 MPa. Dolomite aggregates were used exclusively in the preparation of the reference mixture (M1). M2 and M3 were produced using thermally untreated steel slag at replacement ratios of 50 weight percent (wt%) and 100 weight percent (wt%) for dolomite coarse aggregate, respectively. M4 and M5 were prepared with thermally treated steel slag at the same ratios of replacement as mentioned previously. All the specimens were put into furnaces and heated to 400 °C and 800 °C. In mixtures, M4 and M5 slag were heated to 1000 °C at a rate of 10 °C/min and then kept at a constant temperature for 1 h before cooling naturally to room temperature inside the furnace in order to prevent thermal shock of the slag. The proportioning of mixtures is given in Table 5.

**Table 5.** Mix proportions of concrete.

| Component (kg for 1 m$^3$) | M1 | M2 | M3 | M4 | M5 |
|---|---|---|---|---|---|
| Cement | 380 | 380 | 380 | 380 | 380 |
| Water | 218 | 223 | 228 | 223 | 228 |
| Natural fine Aggregate | 820 | 896 | 966 | 896 | 966 |
| Natural coarse aggregate | 960 | 480 | - | 480 | - |
| Thermally untreated steel slag coarse aggregate | - | 507 | 1014 | - | - |
| Thermally treated steel slag coarse aggregate | - | - | - | 507 | 1014 |

*2.3. Tests Methods*

2.3.1. Tests on Fresh Concrete

The properties of all concrete mixtures in their fresh state are given in Table 6. All testing of fresh concrete was performed according to European standards; density was measured according to EN 12350-6 [36], air content according to EN 12350-7 [37], and consistency according to BS EN 933-2 [30]. The higher specific gravity of steel slag compared to natural aggregates is the cause of the increasing density for mixtures M3 and M5. Figure 5 shows the fresh state tests.

**Table 6.** Properties of fresh concrete.

| Properties | M1 | M2 | M3 | M4 | M5 |
|---|---|---|---|---|---|
| Density (kg/m$^3$) | 2390 | 2493 | 2579 | 2488 | 2588 |
| Slump (mm) | 49 | 48 | 48 | 47 | 48 |
| Air content (%) | 1.8 | 1.9 | 1.7 | 1.9 | 1.9 |

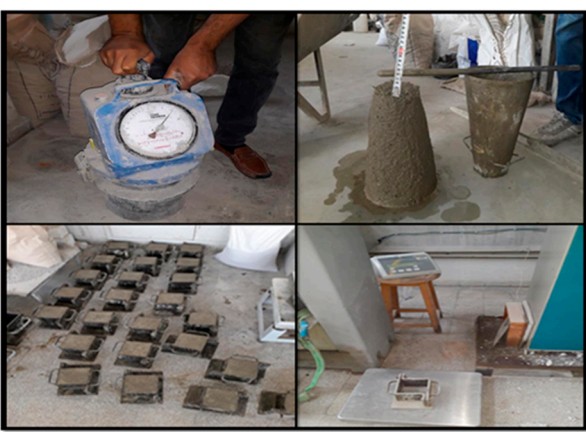

**Figure 5.** Tests on fresh concrete.

### 2.3.2. Tests on Hardened Concrete

- Compressive Strength

Tests on cubic specimens with edge lengths of 100 mm were conducted for compressive strength. Following 24 h of casting, all specimens were de-molded and kept in a water tank at a temperature of roughly $21 \pm 2$ °C for an additional 27 days. At the age of 28 days, the specimens were separated into three groups; the first group was a direct test to determine compressive strength, and the second and third groups were exposed to high temperatures of 400 °C and 800 °C, respectively. The specimens were heated to temperatures between 400 °C and 800 °C at a heating rate of 5 °C/min. The samples were kept at a constant temperature for an hour after reaching the predetermined temperature. The furnace was then turned off and the samples were allowed to cool to prevent thermal shock. The specimens were tested at a loading rate of 0.25 MPa/s according to the BS EN 12390-3 standard [38]. Figure 6 shows the compressive strength test.

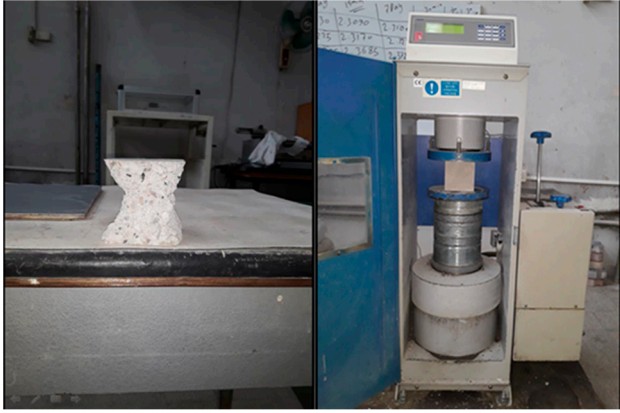

**Figure 6.** Compressive strength test.

- Splitting Strength

Testing for splitting strength was performed on cylindrical specimens that were $75 \times 150$ mm in size. The same techniques as before were employed. The specimens

were tested at a loading rate of 0.03 MPa/s according to BS EN 12390-6 [39]. Figure 7 shows the splitting strength test.

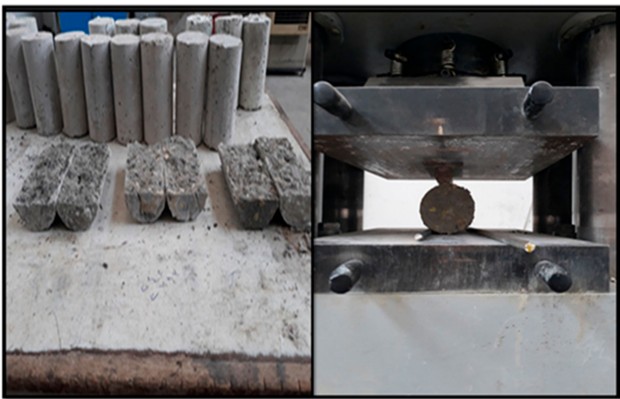

**Figure 7.** Splitting strength test.

## 3. Results and Discussion

Analyzing the properties of fresh mixtures shows that increasing the percentage of slag in mixtures leads to an increase in density. This is due to the high specific gravity of steel slag. Per contra, no significant difference was observed for air content and slump because all mixtures were designed to slump equally to 50 mm and contain air content equal to 2%.

The effects of high temperature on the remaining properties of the concrete mixture were used to assess the fire resistance of the concrete. The residual value of each property is expressed as the percentage change in an observed property's value from its initial value in reaction to an increase in temperature.

### 3.1. Mass Loss

Figure 8 shows the mass loss of the concrete specimens in response to heat, which is represented as a percentage of the initial weight. All concrete mixtures, including the reference mixture, lost about 8% of their weight at temperatures of up to 400 °C. The reference mixture lost more weight (21.15%) at 800 °C than the other mixtures did. This may be caused by the decomposition of calcium carbonate in lime and carbon dioxide [25]. For the M5 mixture, where the lowest loss in weight was obtained (11.41%), the effect of heat treatment on weight stability was evident.

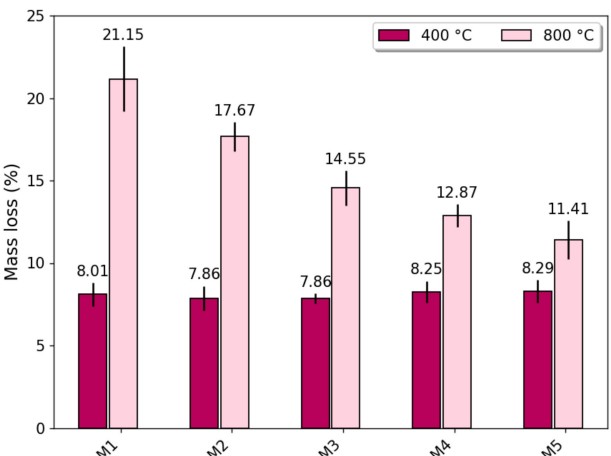

**Figure 8.** Mass loss for concrete mixtures exposed to high temperature.

### 3.2. Compressive Strength

Figure 9 shows the change in compressive strength for mixtures exposed to high temperatures. The M5 mixture, which was entirely made of thermally treated slag, exhibits higher compressive strength (18 MPa) than other mixtures at 800 °C. This is the result of the heat-induced crystallization and mineralization of slag. The least effective mixture was M3, which was produced using slag that had not undergone any thermal treatment. The M3 mixture performs better than other mixtures (33.31 MPa) at 400 °C. This might be as a result of slag expanding, which ensures better filling of porosity in the mixture. At 20 °C, the compressive strength of mixture M1 was extremely close to the design value. However, when the mixture was exposed to high temperatures of 400 °C and 800 °C the compressive strength of the mixture was reduced by 9.6% and 60%, respectively. Results were not consistent with some of the reviewed studies, as those studies showed that the reference mixture was better than steel slag mixtures at a temperature of 800 °C. This may be due to a difference in the properties of the steel slag used, the method of treating this slag, and the duration of exposure concrete samples had to high temperatures [19].

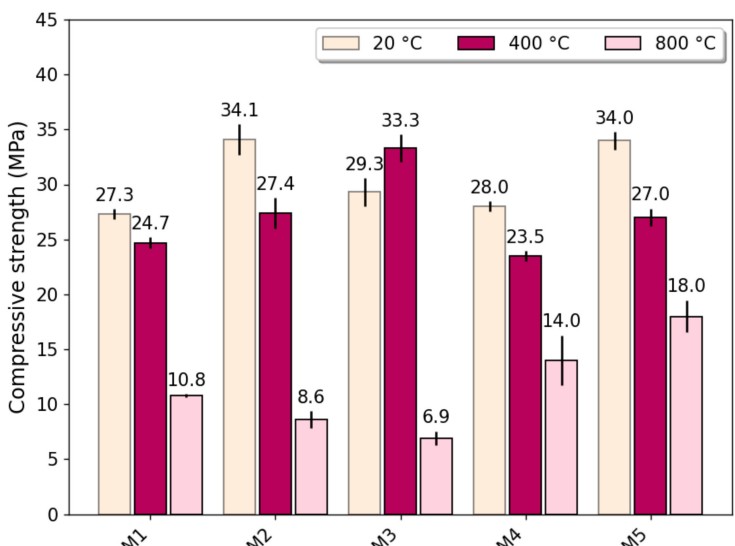

**Figure 9.** Compressive strength for concrete mixtures exposed to high temperature.

### 3.3. Splitting Strength

Figure 10 illustrates the results of splitting tests for mixtures exposed to high temperatures. The results show that the mixtures are weakening overall; for example, mixture M1's strength decreased by 11.2% at 400 °C and by 66% at 800 °C. It is obvious that mixtures M2 and M4 with steel slag as a partial replacement (50%) of natural coarse aggregate provide the best splitting strength compared to other mixtures at room temperature (20 °C). This might be because aggregates are available in a variety of particle sizes and shapes, which helps fill spaces more effectively. The thermally untreated steel slag-containing M2 and M3 mixtures perform better than other mixtures at 400 °C. This might be a result of the expansion that occurred in the steel slag aggregate, which resulted in better stress distribution. Regardless of the type of aggregates used, all mixtures had a significant loss in strength at 800 °C.

The evolution of the mechanical properties with mass loss is shown in Figure 11. It is evident that there is a relationship between the loss of mass and the deterioration of compression resistance. The most degraded material is M3, incorporating 50% heat-treated aggregates (Figure 11a). Additionally, the relationship between the loss of mass and tensile strength is significantly less than compressive strength, particularly for materials heated to 800 °C (Figure 11b). For all materials studied, Figure 12 shows the evolution of tensile strength as a function of compressive strength. Figure 12a shows that, regardless of the type

of aggregates used, an analytical model of the power type that approaches the equation of Eurocode 2 cannot be used to estimate the tensile strength of materials heated to 800 °C. However, a linear model (Figure 12b) is more adequate. All materials heated to 800 °C, with the exception of the reference and M5, are basically at the bottom limit of the ±30% interval.

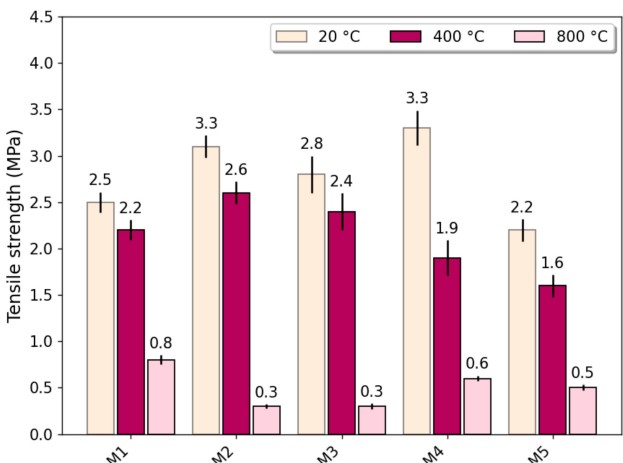

**Figure 10.** Splitting tensile strength for concrete mixtures exposed to high temperature.

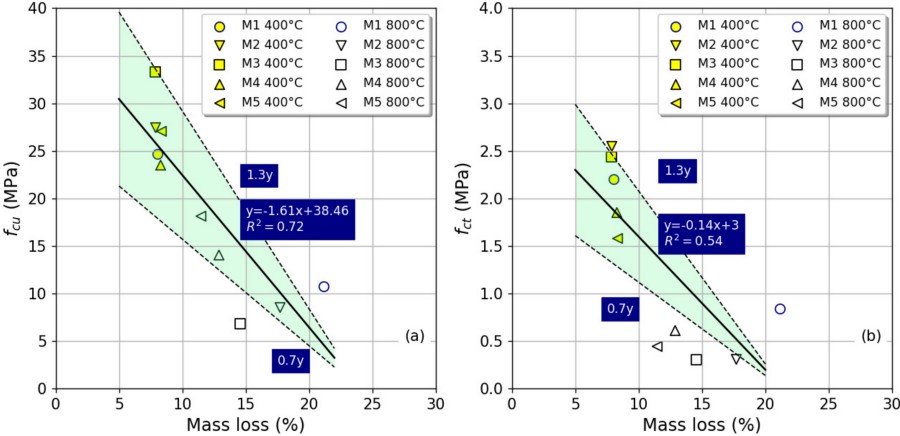

**Figure 11.** Evolution of mechanical properties with mass loss: (**a**) compressive strength, (**b**) tensile strength.

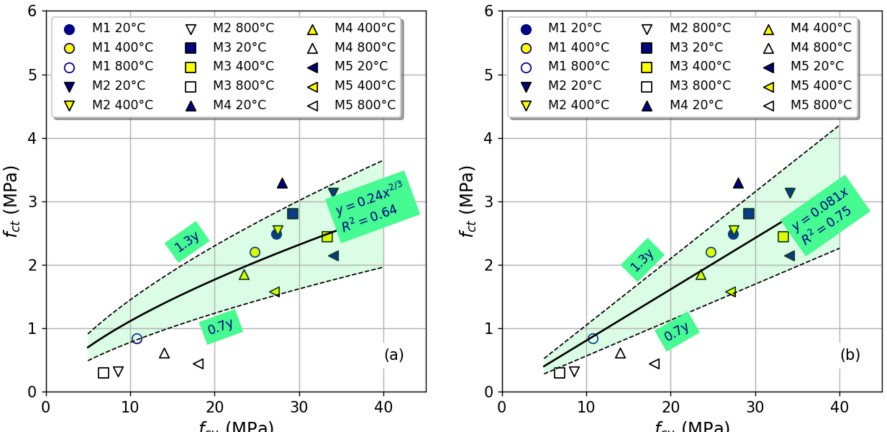

**Figure 12.** Relationships between compressive and tensile strengths.

### 3.4. Microstructure of the Studied Concretes

An optical microscope with a magnification ratio of 100 was used to examine the difference in the microstructure of mixtures before and after exposure to high temperatures in order to clarify the results. Figure 13 presents the results. For all mixtures, there were no cracks visible at room temperature, while for mixtures M2 and M3 relatively tiny cracks at the interfacial transition zone (ITZ) were seen at 400 °C. For other mixes, there are no visible cracks. Large cracks were observed in the thermally untreated slag's structure, as well as in mixtures M1, M2, and M3 inside the ITZ at 800 °C. However, compared to the other mixtures, M4 and M5 showed smaller cracks. This explains why, after being exposed to high temperatures, mixture M5 performed the best in terms of compression.

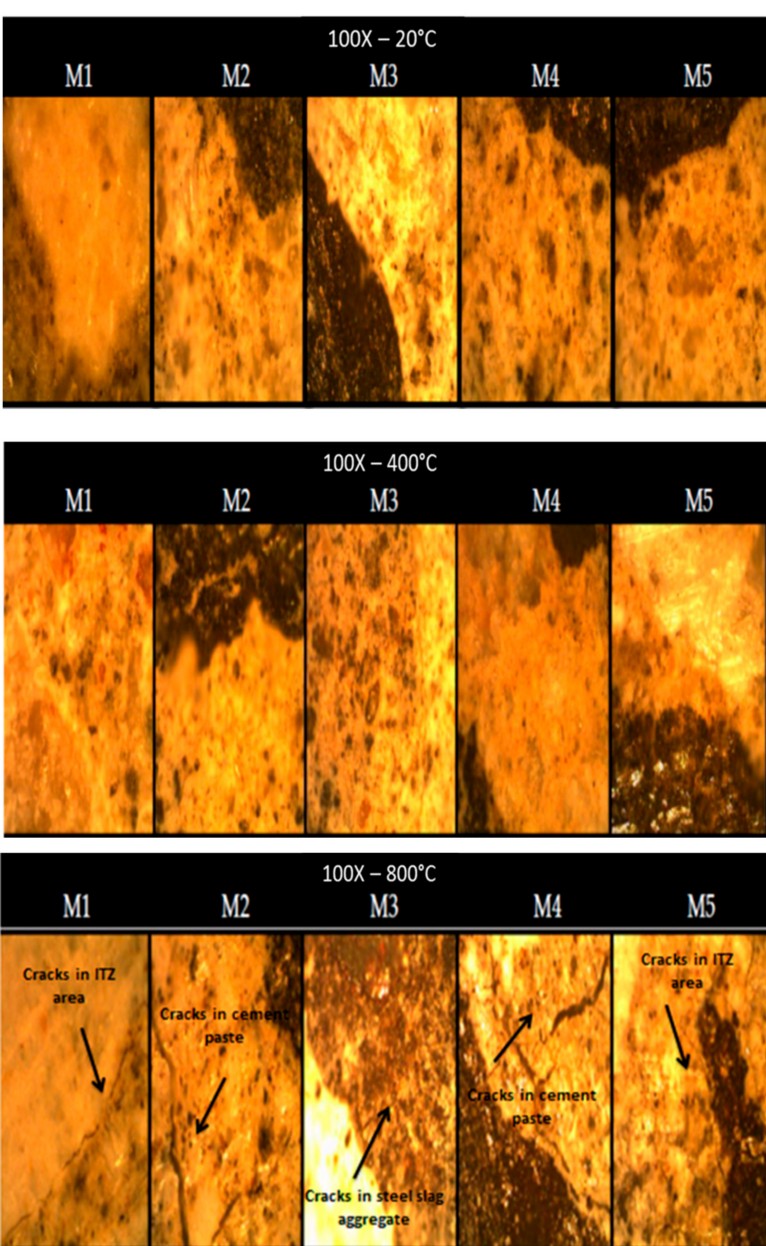

**Figure 13.** The microstructure of concrete before and after exposure to high temperature.

As shown in Figure 9, when mixtures M2 and M4 (with partial replacement ratios) were compared to mixtures M3 and M5 (with an entire replacement ratio of steel slag aggregate), the mixtures with fully steel slag had significantly higher compression strength. Large cracks in the cement paste were caused by additional stresses induced by the differ-

ence in thermal expansion coefficients between steel slag aggregate and natural aggregate, as shown by the microstructure images in Figure 13. The internal cracks in the thermally untreated steel slag's internal structure (Figure 14) are the reason why mixture M5 outperformed mixture M3 with thermally untreated steel slag aggregate.

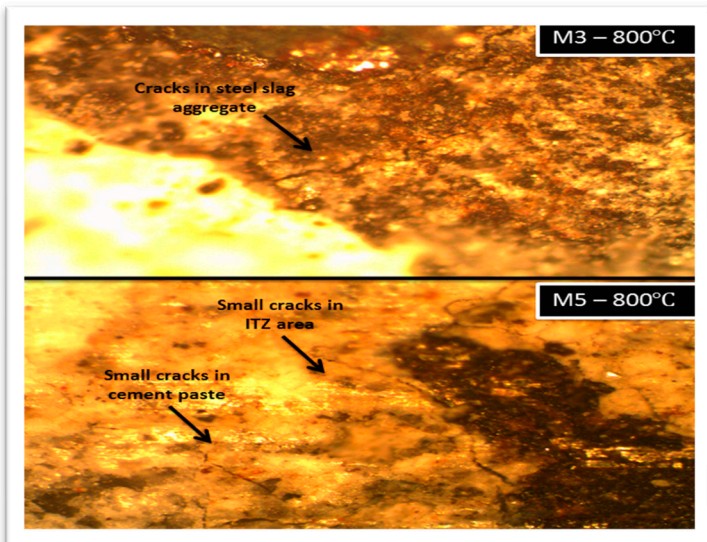

**Figure 14.** The microstructure of mixtures M3 and M5 at 800 °C.

The crack widths in many mixes were estimated using the image analysis software Image J Version 1.54f. The crack widths for the M1 reference mixture in the ITZ area were estimated to be 20 µm at 800 °C, whereas the crack widths for the M5 mixture were 10 µm. For the M2 and M4 mixtures, the cement paste cracks were up to 10 and 25 µm, respectively. While the thermally treated aggregate was stable and free of cracks, the thermally untreated steel slag aggregate had tiny cracks visible in the aggregate's bulk. As calcium hydroxide is formed in the unheated steel slag as a result of the interaction of free calcium oxide with water when it is stored in yards around factories, it can be claimed that the thermal treatment of steel slag aggregate contributes to the stability of these aggregates. Since calcium hydroxide dissolves at temperatures lower than 500 °C, these aggregates become unstable when used in concrete that is exposed to high temperatures. These aggregates are significantly improved by thermal treatment, which turns them into a component that enhances the mechanical qualities of refractory concrete.

## 4. Conclusions

In this study, the properties of refractory concrete made with thermally treated and untreated steel slag were compared. The results can be summarized as follows:

1.  The density of fresh concrete mixtures containing slag increased with increasing slag percentage compared to the reference mixture. Conversely, there was no significant difference in air content and slump between the reference mixture and others containing steel slag.
2.  In comparison to mixtures containing natural aggregate, mixtures including steel slag are denser.
3.  Thermally treated steel slag has a significant effect on the compressive strength of concrete exposed to high temperatures of 800 °C when used to replace 100% of natural coarse aggregate.
4.  A mixture containing fully thermally treated steel slag as a coarse aggregate after being exposed to a high temperature of 800 °C demonstrated the effect of thermal treatment on the stability of the weight of concrete.
5.  At 800 °C, the thermally treated steel slag-containing mixtures' cracks were smaller than those of the other mixtures.

6. The mixtures that included steel slag to replace part of the natural coarse aggregate had the highest splitting tensile strength.

7. Results showed that thermally treating slag could increase the fire resistance of concrete exposed to high temperatures up to 800 °C.

According to the results of this research, we recommend that that steel slag be thermally treated before its use as a partial or complete alternative to natural coarse aggregate.

**Author Contributions:** Conceptualization, formal analysis, investigation, and the performing of tests: M.A. (Munaf Alkhedr); supervision and validation of test results: M.A. (Majed Asaad) and M.I.; review and editing of the original manuscript: G.W. All authors have read and agreed to the published version of the manuscript.

**Funding:** This research received no external funding.

**Data Availability Statement:** The data that support the findings of this study are available from the first author (Munaf Alkhedr) upon reasonable request.

**Conflicts of Interest:** The authors declare no conflict of interest.

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
