# Peer review of "Experimental Study on the Physical and Mechanical Characteristics of Refractory Concrete Using Heat-Treated Steel Slag Coarse Aggregates"

_infrastructures, doi:10.3390/infrastructures8100151_

Round 1

Reviewer 1 Report

Work done is original and experiments have been planned in a technical way. The references are sufficient and correctly used. The results are properly discussed. However, some minor modifications of manuscript are recommended at several places in a more precise form before being accepted for publication. Minor revisions in the light of comments/suggestions are listed.

Abstract
1. The Abstract should be rewritten by summarizing the research results obtained in this paper.

Keywords

2. Adopt upper case initial letter for all words.

Introduction

1. The paper contains omissions in the description of the experimental program. It should be properly summarized in the end of the Introduction part, by introducing all the test carried out on the specimens.

Materials and Methods

1. The term of "table 1, 2,…." should be replaced by adopting the upper case of initial letter "Table 1, 2,…" in all the text.

2. In Table 5 & Table 6, the unit of (kg/m3), must be corrected and replaced by (kg/m3).

3. In Paragraph 2.3.1: The testing devices used for fresh properties must be mentioned in the text.

4. Supported references for all physico-mechanical testing Standards cited in the text should be added.

Results and Discussion

 1. The degradation of the microstructure of specimen after heating can be easily analyzed by the dynamic elastic modulus-measurements using an Ultrasonic Pulse Velocity tester (UPV). Why the authors omitted to use this non-destructive test?

2. To perform a better mechanical strength for specimen-based heat-treated steel slag aggregates, the SEM analyser system coupled with Energy Dispersive X-Ray spectroscopy (EDX) can be used. These analyses may be effective to evaluate the effect of heat-treated aggregates on the development of hydration components of used binder.

3. It’s well known that the slag aggregates exhibit a roughen surface, which makes the specimen more adhesiveness. Why the authors did not explore this issue after heating treatment, through the SEM Micrographs analyses?

Conclusion

1. The conclusion is a series of summary points without any reflection. The limitations of the study should be discussed ;.

Finish.

Quality of English Language is acceptable, despite the proposed of minor modifications.

Author Response

The authors would like to thank reviewer 1 for his remarks which led to improving the quality of the draft. All required modifications were highlighted in yellow in the revised manuscript.

Reviewer 2 Report

The theme of the paper is interesting, but the text needs to be better designed and formatted.

Tables and figures are presented outside the margins; there are blank lines between paragraphs; spaces between titles and texts do not follow a formatting rule; the figures need to be of better quality (especially figures 5 and 6); figure 4 does not exist.

The abstract must contain only one paragraph.

A paragraph is a set of sentences and must contain the development of an idea.

The objective of the work already states a conclusion ‘in order to use it as a coarse aggregate in concrete, with a positive effect on the mechanical properties of refractory concrete.” The objective should be rewritten with a focus on the contribution of research to advancing the frontier of knowledge.

In the Introduction: highlight the problems related to slag use as aggregate (instability) and the importance of its aging.

Methodology texts that explain/describe trials are not necessary.

It is necessary to describe the treatment of slag.

It is necessary to specify the equipment and parameters used in all tests.

Figure 5 requires the identification of compounds as in Figure 6.

It would be interesting to present the results of tests on fresh concrete in the Results and discussion item.

The discussion of compressive strength and splitting strength results must be literature-based.

There is no such thing as 100x100x100 in scientific language.

The statement “It is evident that there is a relationship between the loss of mass and the deterioration of compression resistance.” is not supported by the data.

Indicate the cracks observed in the microstructures in Figure 12.

Regarding the statement “For all mixtures, there were no cracks visible at room temperature.”: was the microstructure of the entire sample analyzed?

Author Response

The authors would like to thank reviewer 2 for his remarks which led to improving the quality of the draft. All required modifications were highlighted in yellow in the revised manuscript.

Reviewer 3 Report

This article is interesting and provides useful information on “Experimental Study on the Physical and Mechanical Characteristics of Refractory Concrete Using Heat-Treated Steel Slag Coarse Aggregates” Therefore, the paper and its results are interesting, and the paper is considerably well written, so that I consider that a good paper for a high level Journal as Infrastructures can be obtained based on this first version, provided that some issues are corrected and improved according to the following indications.

The highlights are useful and the abstract is well structured and summarizes the paper

Introduction section presents an interesting state of the art, relating the previous references with the work in the paper. However, few references are indicated. There are interesting studies on topics of substitution of aggregates for other and unreferenced materials, for example: “Strength Performance of Different Mortars Doped Using Olive Stones as Lightweight Aggregate”, 2022, Buildings 12(10),1668…. Methodology for the environmental analysis of mortar doped with crumb rubber from end-of-life tires”, 2023, Construction and Building Materials 399,1325192….

Methodology section presents very interesting content, but I consider that this section must be reorganized to present in a clearer way the information. Figure 1, 5, 6, 10 y 11 doesn't look very good. I recommend improving the quality. Figure 2 has no scientific value.

In Table 2, Specific Gravity has no units. Table 3 restructure the size of the columns to better read the content

Conclusions section needs a restructuring so that the information is transmitted to the reader in a clearer way. Expand conclusions. I suggest that conclusions to include some recommendations for specialists in the field, based on results of research.

Author Response

The authors would like to thank reviewer 3 for his remarks which led to improving the quality of the draft. All required modifications were highlighted in yellow in the revised manuscript.

Round 2

Reviewer 2 Report

The authors have sufficiently addressed the comments given earlier. However, I will suggest the authors perform a thorough proof check of the document before its acceptance.

Reviewer 3 Report

The authors have taken into account all my suggestions, and the paper has improved considerably due to this changes. The explanation of the changes is also clear and rigurous.

Therefore, in my opinion the paper is now worthy of publication.